# Fermentation for Designing Innovative Plant-Based Meat and Dairy Alternatives

**DOI:** 10.3390/foods12051005

**Published:** 2023-02-27

**Authors:** Fatma Boukid, Abdo Hassoun, Ahmed Zouari, Mehmet Çağlar Tülbek, Marina Mefleh, Abderrahmane Aït-Kaddour, Massimo Castellari

**Affiliations:** 1ClonBio Group Ltd., 6 Fitzwilliam Pl, D02 XE61 Dublin, Ireland; 2Univ. Littoral Côte d’Opale, UMRt 1158 BioEcoAgro, USC ANSES, INRAe, Univ. Artois, Univ. Lille, Univ. Picardie Jules Verne, Univ. Liège, Junia, F-62200 Boulogne-sur-Mer, France; 3Sustainable AgriFoodtech Innovation & Research (SAFIR), F-62000 Arras, France; 4LRGP, UMR 7274 CNRS—Université de Lorraine, 2 Avenue de La Forêt de Haye TSA, 40602, F-54518 VANDŒUVRE, France; 5Saskatchewan Food Industry Development Centre, Saskatoon, SK S7M 5V1, Canada; 6Department of Soil, Plant and Food Science (DISSPA), University of Bari Aldo Moro, 70126 Bari, Italy; 7Université Clermont Auvergne, INRAE, VetAgro Sup, UMRF, F-63370 Lempdes, France; 8Institute of Agriculture and Food Research and Technology (IRTA), Food Industry Area, Finca Camps i Armet s/n, 17121 Monells, Spain

**Keywords:** plant proteins, precision fermentation, food innovation, safety, health and nutrition, digitalization

## Abstract

Fermentation was traditionally used all over the world, having the preservation of plant and animal foods as a primary role. Owing to the rise of dairy and meat alternatives, fermentation is booming as an effective technology to improve the sensory, nutritional, and functional profiles of the new generation of plant-based products. This article intends to review the market landscape of fermented plant-based products with a focus on dairy and meat alternatives. Fermentation contributes to improving the organoleptic properties and nutritional profile of dairy and meat alternatives. Precision fermentation provides more opportunities for plant-based meat and dairy manufacturers to deliver a meat/dairy-like experience. Seizing the opportunities that the progress of digitalization is offering would boost the production of high-value ingredients such as enzymes, fats, proteins, and vitamins. Innovative technologies such as 3D printing could be an effective post-processing solution following fermentation in order to mimic the structure and texture of conventional products.

## 1. Introduction

Since the Neolithic times, fermentation has been a natural process passed down through generations to produce several types of foods and beverages. Fermentation was primarily used for food preservation and shelf-life extension. Nowadays, there are a large variety of fermented animal and plant products made using a wide range of raw materials, microorganisms, and manufacturing techniques [1]. Fermented meat products such as salami, ham, and sausages have traditionally been produced all around the world and currently occupy a special position in the gastro-economic trade of meat products [2,3]. Fermented dairy products including cheese and yoghurt are staples known to contain potentially probiotic microorganisms such as lactic acid bacteria [4,5]. Fermentation has also been applied to plant-based sources such as coffee, bread, chocolate, wine, and olives in order to improve their nutritional value, aroma and taste, texture, and stability [6,7]. Overall, fermented foods are a crucial part of the human diet owing to their health benefits and particular flavor, aroma, and texture [8].

As a part of the shift towards plant-based diets and alternative proteins, dairy and meat alternatives have become increasingly popular around the world. Cereals, legumes, oil seeds, nuts, and vegetables are the main sources used to make alternative products [9]. These plant-based sources have lower nutritional quality than animal proteins and might induce undesirable flavors. Furthermore, plant proteins have different compositions and structures than animal proteins resulting in different functional features, including solubility, gelling, emulsifying, and foaming [10,11]. Thus, one-to-one replacement of animal proteins with plant proteins to achieve a similar texture and mouthfeel to that of traditional products could be challenging. Among the strategies to recreate a meat/dairy-like experience, manufacturers rely on the use of additives such as fats, starches, flavorings, colorings, and stabilizers [12,13]. Several plant-based products are perceived to be less nutritious (i.e., high in sugar, salt, and fat and low in protein contents) compared to animal products. Furthermore, the excessive use of additives is not appreciated by many consumers seeking natural and clean-labelled ingredients. Alternatively, manufacturers rely on processing for protein functionalization, for example, of textured vegetable proteins, to create a meat-like fibrous structure [14]. However, most of the resulting products could be classified as ultra-processed foods, i.e., the group 4 of NOVA classification (a classification system that groups all foods according to the nature, extent, and purposes of the industrial processes they undergo) [15]. The healthiness of ultra-processed foods is a controversial debate because of potential health issues such as cancer, obesity, and cardiovascular diseases [16,17].

Therefore, fermentation has been playing a pivotal role in recent years to make plant-based alternatives to meat/dairy with minimal additives/processing (Figure 1). There are three types of fermentation used in plant-based products. Traditional fermentation intends to produce traditional foods and beverages. Various microorganisms (mainly bacteria and yeasts) are involved in the fermentation of plant-based raw materials [6]. These microorganisms may be indigenously present on the substrate or added as a starter culture, or they may be present in the ingredient(s) [18]. Depending on the raw material properties, the used microorganism, process conditions and substrate composition, fermentation can induce several changes in the organoleptic and nutritional properties of fermented plant-based ingredients/foods. Several pieces of recent research have underlined the potential of fermentation to improve the sensory and nutritional quality of plant-based fermented products [19,20,21]. Biomass fermentation aims to obtain single cell proteins (SCP). Recently, precision fermentation emerged as a novel targeted technology for food applications with the aim of providing high-value compounds [22]. This novel technology leverages metabolic engineering tools to serve as a factory of ingredients such as protein, pigments, vitamins, and fats to upgrade the quality of plant-based alternatives [23,24]. Furthermore, fermentation is important for sustainable food solutions and can provide a positive impact on the sustainability index for the food industry [25,26,27]. Compared to conventional protein sources, biomass fermentation for protein production can rapidly produce a very high protein content among other nutrients. Furthermore, food by-products and wastes can be used as substrates to be transformed in high-value food and feed products [28,29,30,31]. This is an environmentally friendly strategy that could encourage a more energy-efficient and sustainable economy.

Within this framework, this review aims to provide a better understanding of the opportunities and challenges related to the use of fermentation for developing plant-based products. Firstly, the market landscape and segmentation of fermented plant-based products is addressed with a focus on dairy and meat alternatives. Secondly, the impact of fermentation on the sensory and nutritional characteristics of plant-based dairy and meat products is discussed. Finally, a section is dedicated to capture the role of digitalization in facing the current and forthcoming challenges of fermented plant-based foods.

## 2. Market Landscape of Conventional and Innovative Plant-Based Foods Made Using Fermentation

The global market of plant-based products made using fermentation was valued at USD 329.29 million in 2021 and is expected to reach USD 422.26 million by 2026, with a compound annual growth rate (CAGR) of 5.0% [32]. Figure 2 illustrates the evolution of the number of the products launched in the global market during the last two decades (2002–2022). From 2002 to 2012, the number of fermented plant-based products increased, but at a low speed. During the period 2013–2022, this market witnessed an exponential growth and reached a peak in 2021 owing to the expansion of fermented plant-based meat and dairy alternatives [33], and it is expected to keep growing in the forthcoming years. The outbreak of COVID-19 contributed to fueling consumer interest in fermented plant-based products, owing to their health benefits such as boosting immune system performance and ameliorating gut health and inflammatory responses [34,35].

The global fermented plant-based market is highly fragmented, including conventional (e.g., bakery) and emerging (i.e., alternative to meat and dairy) products. Overall, bakery is the largest fragment, followed by dairy alternatives, sauces and seasonings, meat alternatives, and vegetables, as well as ready-to-eat meals (Table 1). Europe holds the largest share of the market of fermented plant-based products, with a 73% market share, followed by Asia-Pacific, North America, Latin America, the Middle East, and Africa (Table 2). Europe generated the highest revenue of USD 96.99 million in 2020, which is projected to reach USD 129.47 million by 2026 [36]. In 2020, consumer interest in vegan (1.9%), vegetarian (3.1%), and flexitarian (57.1%) foods increased across Europe [37].

Plant-based dairy and meat alternatives produced by fermentation account for 26.30% and 8.56%, respectively, of the total fermented plant-based foods sold on the market (Table 1). There is now a wide range of commercial plant-based dairy products. Even though plant-based “yoghurt” is still at an early stage, it has registered the highest demand in the market of fermented plant-based foods [38]. Manufacturers have focused on diversifying their product portfolio through broadening their range of flavors, while using health-beneficial probiotic bacteria [39,40]. The global plant-based “yoghurt“ market was estimated at USD 2.02 billion in 2020, and it is expected to achieve significant growth in the coming years [32]. Europe dominates ~50% of this market share [38]. Regarding the fermented “cheese“ market, the global market was valued at USD 2.70 million in 2019 and is expected to reach USD 4.58 billion by 2025, at a CAGR of 8.91% [41]. Fermented plant-based “cheese“ is available in different types (e.g., camembert, Roquefort, and feta), forms (shreds, blocks, and slices), and textures (hard, soft, creamy, and spreadable) to fit different uses [12]. This industry is increasingly relying on *fermentation* technologies to produce high-quality products while reducing the use of starches and vegetable fats [42].

Plant-based fermented dairy products are generally made by fermentation of aqueous extracts obtained from different raw plant-based materials [43,44]. Based on Mintel GNPD, the most used fermented ingredients derive from coconut (n = 575), soy (n = 570), oat (n = 143), rice (n = 73), wheat (n = 55), nut (n = 42), bean (n = 27), and almond (n = 27). In most cases, the used ferments are not reported on the products’ labels, apart from lactic acid bacteria (n = 409) and vegan bacteria culture (n = 10). Lactic acid fermentation of plant-based products is commonly applied for improving product palatability and nutritional and health-promoting quality [6,20,45]. Products such as tempeh (fermented soybean) and fermented tofu were not included because they were not originally designed as meat alternatives [46].

Recently, the use of precision fermentation is emerging to unlock the potential of alternative products through providing targeted ingredients with particular features to deliver a unique dairy/meat-like experience. According to GFI database (https://gfi.org/resource/alternative-protein-company-database/ accessed on 20 September 2022), 151 startups use fermentation in developing plant-based alternatives, where precision fermentation (n = 68, 45%) attracts the highest interest, followed by fermentation to obtain protein-rich biomass (n = 60, 40%) and classic fermentation (n = 23, 15%). These startups produce fermented ingredients (n = 60) as well as dairy (n = 59), meat (n = 138), seafood (n = 32), and egg (n = 13) alternatives. Most of these products are not available on supermarket shelves and are still under development.

## 3. Impact of Fermentation on the Quality of Plant Based-Dairy Alternatives

### 3.1. Plant-Based Beverages (Milk Alternatives)

Plant-based beverages are water-soluble extracts of legumes, oil seeds, nuts, cereals, or pseudocereals, and have a similar appearance and consistency to milk [47,48]. To date, there is no consensus definition and/or classification of plant-based beverages/drinks, and these are also referred to as alternatives/substitutes to milk/dairy or plant-based milks or dairy. Categorizing plant-based beverages as cow’s milk substitutes is still debatable. Raw milk has been defined by the Commission Regulation (of the EU) No 605/2010 as milk secreted from the mammary glands such as animals and humans. Thus, plant-based beverages do not meet the definition of milk. Furthermore, several studies reported that plant-based sources are not nutritionally equivalent to milk [49,50,51]. Therefore, plant-based beverages can be considered as a separate category offering fermented/non-fermented options to consumers. The main plant sources used for making plant-based beverages are soy, almond, coconut, oat, and rice [19,49,52,53]. Due to the increased demand for this category, there is room for new sources such as lentils, peanuts, quinoa, lupin, and peas [25,54,55,56]. The common process for manufacturing non-fermented beverages includes wet milling, filtration, formulation (the addition of ingredients), sterilization, homogenization, emulsification, and storage. This process can be adjusted by adding/replacing some operations to fit the specific features of the raw materials and avoid the deterioration of product quality [6,53].

The process of fermented plant-based beverages follows the same steps of the non-fermented variants, but with the addition of two supplementary steps, conditioning (to reach the optimal temperature for the growth of the microorganisms) and fermentation (under specific conditions suitable for the used microorganism(s)) [19,54]. Plant proteins has been shown to be efficient carriers of probiotics [57]. Most plant-based beverages are fermented using lactic acid bacteria, mainly *Lactobacillus* (*Lactobacillus* spp., *L. casei*, *L. helveticus*, *L. fermentum*, *L. reuteri*, *L. acidophilus*, *L. rhamnosus* and *L. johnsonii*), *Bifidobacterium* (*Bifidobacterium animalis* ssp. *Lactis*), *Streptococcus* (*Streptococcus thermophilus*), and *Enterococcus* (*Enterococcus faecium*) [25,55,58,59,60]. Plant-based substrates favor high viability of fermenting microorganisms, and consequently result in probiotic dairy-free products (>10^6^ CFU/mL of lactic acid bacteria cocci) [57,58,61]. The duration of fermentation of plant-based beverages is typically ≈12–24 h, depending on the raw plant-based material properties, the used microorganism(s) and the final product features [62,63]. Excessive fermentation time (>24 h) may result in the formation of undesirable compounds that can negatively impact the nutritional and organoleptic qualities [6,25,64].

The use of fermentation enhanced the nutritional value and palatability of plant-based beverages. Pectinases released during lactic fermentation by *Lactobacillus* and *Streptococcus* improved the content of proteins, the amino acid profile, and protein digestibility [53,59,65]. It also contributed to the formation of peptides with bioactive activities, e.g., ACE-inhibitory, antioxidative, and anti-microbial activities [59,66]. Starters such as *Lactobacillus* and *Bifidobacteria* enhanced minerals content and availability as well as antioxidant properties in beverages made with soy, chickpea, and red bean [6,67,68]. In addition, fermentation promoted an increase in organic acid and short-chain fatty acids concentrations, which in turn enhanced the absorption and solubility of minerals (calcium, iron, and zinc) and vitamins [52,69,70,71,72]. Lactic bacteria could synthesize vitamins that are naturally lacking or absent in plant sources, such as vitamin B and K [6,69,73]. Additionally, the activity of β-glucosidase increased the isoflavone content, which contributed to enhancing the digestion of plant-based beverages [73,74,75]. Lactic bacteria were found to be efficient in mitigating antinutrients such as stachyose, raffinose, phytate, oligosaccharides, tannins and protease inhibitors, which in turn increased the bioavailability of minerals [25,76]. Similarly, raffinose was reduced by 60% in fermented moringa leaves and beetroot extract with *Lactobacillus plantarum* and *Enterococcus hirae* [77]. Furthermore, fermentation increased antibacterial activity against *Bacillus cereus*, *Escherichia coli*, *Listeria monocytogenes* and *Staphylococcus aureus* [77]. Radical scavenging activity and phenolic content as well as minerals (calcium an iron) improved after fermentation. *C. vulgaris* and soy extract fermented using *Lactobacillus fermentum* and *Lactobacillus rhamnosus* resulted in improved polyphenol content and dietary antioxidant capacity compared to fermented soy extract [78]. Fermentation mitigated plant protein allergens resulted in the loss of the IgE-binding ability of epitopes [9,79]. For instance, conglycinin (7S) and glycinin (11S) were found to be drastically reduced in soymilk fermented by *Enterococcus faecalis VB43* [80]. In addition, *Lactiplantibacillus plantarum* subsp. *plantarum* reduced allergens (β-conglycinin and glycinin) in chickpea-based beverages [81].

The beany flavor related to *n*-hexanal and *n*-hexanol, acetate, isovalerate, and 2-methylbutyrate was dramatically reduced by *Bacillus subtilis, Lactobacillus* species and edible fungi (e.g., *Naematelia aurantialba, Lycoperdon pyriforme, Phellinus igniarius* and *Agrocybe cylindracea*) [82,83,84,85]. This resulted in reduced bitterness in fermented soymilk and legumes beverages [82,83,84,86,87]. On the other hand, fermentation favored the formation of desirable volatile flavors such as acetoin, diacetyl (2,3-butanedione), and acetaldehyde, depending on the used microorganism(s) [64,85,88]. For instance, new and pleasant aromatic notes were perceived after fermenting lupin, such as cheesy aromas (using *Lb. reuteri*), fatty aromas (using *Lb. brevis*, *Lb. delbrueckii*), and roasted aromas (using *Lb. amilolyticus*, *Lb. helveticus*) [89]. For texture, no thickening effect is required in milk-like beverages, and thus fermentation is primarily used to improve taste and flavor [52,90,91]. Therefore, the key selection criterion of plant materials is their solubility, in order to avoid powdery mouthfeel [91]. The formation of antimicrobial compounds against e.g., *Bacillus cereus*, *Staphylococcus aureus*, and *Pseudomonas aeruginosa* can contribute to extending shelf-life [92].

### 3.2. Spoonable Yoghurt-Like Products

Plant-based yoghurt-like products are generally made by fermenting aqueous extracts obtained from oat, pea, cashew, almond, coconut, and soy [93,94,95,96,97]. Unlike fermented beverages, the major challenge of plant-based “yoghurt” alternatives is recreating the viscous texture of dairy yoghurt. For this reason, commercial non-fermented plant-based “yoghurt” alternatives rely chiefly on the use of thickening agents (e.g., natural gums, proteins, starches, pectin, and agar) to reach the desired consistency and ensure product stability [43,98,99]. However, adding additives might negatively impact the consumers’ acceptance, as they might prefer clean-labelled products. In fermented plant-based yoghurt, *S. thermophilus* and *L*. *delbrueckii* subsp. *Bulgaricus* strains are the most-used starters, with additional optional species for enhancing the nutritional and/or organoleptic qualities. *S*. *thermophilus* and *L*. *delbrueckii* subsp. *Bulgaricus* were reported to efficiently change oat protein concentrates’ structure to favor their aggregation, which resulted in increased consistency [93]. Fermented oat-based yoghurt products are appreciated for their desirable flavor and texture [100]. It was reported that the use of lactic bacteria producers of exopolysaccharide (e.g., *Weissella confusa*) might improve viscosity and mouthfeel, reaching similar features to those of conventional dairy [101,102]. *Weissella confusa* increased the viscosity and the water-holding capacity of quinoa-based yoghurt [102]. It also improved protein digestibility and resulted in high final viable cell counts (>10^9^ CFU/mL) [101]. Fermenting sprouted tiger nut tubers with Lactobacillus bulgaricus and Streptococcus thermophilus provided probiotic products with increased protein and amino acids contents, improved sensory attributes, and reduced levels of anti-nutritional compounds [103].

### 3.3. Plant-Based “Cheese” Alternatives

Plant-based “cheese” analogues can be subdivided into two categories: those made using fermentation and those not. Non-fermented cheeses are the most available in the market, and they are made using vegetable oils/fats (e.g., sunflower, coconut or palm oils) and polysaccharides (e.g., starches, gums, and fibers) as the main ingredients [104,105,106]. For vegetable oil-based “cheese” alternatives, the selection of the type of fat is crucial for determining the quality of the end product. For instance, fresh and ripened Edam-type “cheeses” prepared using palm oil had a similar fat content and texture to their dairy counterparts [104], while those made of sunflower oil showed high spreadability and low firmness [107]. Overall, the use of fat enables manufacturers to imitate the texture and meltability of dairy cheese, but not the stretchiness and flow [45,108]. Polysaccharides were used in the making of soft “cheeses” such as Mozzarella, and resulted in a low fat content, soft texture, and some semblance of stretching [105,109]. In dairy products, stretchability is related to the weakening of non-covalent casein–casein interactions, which is not possible using plant-based proteins [110,111]. This explains why most fermented commercial “cheese” alternatives have a low protein content, and manufacturers rely chiefly on fats and polysaccharides [110]. A few studies have investigated other matrices, with protein content around 20%. A prototype of cheddar “cheese” made by including 30% of zein showed similar softness, stretchability, and meltability to the dairy type [108]. Soy and cashew-based “cheese” alternatives also showed high protein and low fat contents, while being appreciated for their color and flavor [112].

The use of plant-based fermentation in “cheese” is not yet widespread on the market [113]. The impact of fermentation on plant-based dairy is summarized in Table 3. Plant-based extracts from cashew, soy, and nuts were fermented using lactic bacteria and then used for making “cheese” alternatives [53,114]. A soy “cheese” spread made using lactic bacteria and the addition of glucono-δ-lactone showed improved texture [115]. The use of lactic bacteria and/or *Geotrichum candidum* improved the sensory properties of soy-based “cheese” in its fresh state and after ripening (10 °C for 28 days) [116]. It was also reported that this “cheese” had better features than that made only using lactic bacteria [116]. A prolonged fermentation (7 days) of soy protein isolates with *L. helveticus* strains enabled the formation of cheese flavors (3-methylbutanal, 2-methylbutanal and benzaldehyde). This result might be an opportunity to modulate fermentation time to obtain natural flavoring compounds [117].

More matrices are being explored in the literature. Different pea protein isolates and olive oil emulsions were fermented with a commercial bacterial inoculum (Vega^TM^) [113]. The authors found that optimal fermentation-induced pea protein gels can be produced with 10% protein content and 10% olive oil levels without compromising gel hardness. A peanut extract-based spread was prepared using probiotic microorganism *Lactobacillus rhamnosus* NCDC18 [118]. The product showed acceptable appearance, yet no sensory characterization was performed. Fermenting flaxseed oil cake using a combination of lactic acid bacteria, *Penicillium camemberti*, and *Geotrichum candidum* enabled the manufacture of a camembert-like “cheese“ with improved oil oxidative stability [119]. Fermented rice milk was prepared using lactic acid bacteria and various coagulation agents (gelatin, xanthan gum, or agar). Although gelatin treatment enabled the best sensory scores, its animal origin can limit its use in vegan alternatives [21]. Fermented cashew using quinoa starter inoculum (dominated by *Pediococcus* and *Weissella*) was used to make “cheese” alternatives. The nutritional quality was marginally changed, while allergenicity associated with cashew was drastically reduced. Moreover, a high viable bacterial count was recorded (10^8^–10^9^ CFU/g) [120].

**Table 3 foods-12-01005-t003:** Impact of fermentation on the quality of plant-based products.

Substrate	Starters	Effects	References
Broad bean and chickpea beverages	*Streptococcus thermophilus*, *Lactobacillus delbrueckii* subsp. Bulgaricus, and a mixture of *Lactobacillus casei* and *XPL-1*, which is a mixed culture containing *Lactococcus lactis* subsp. cremoris, *Lactococcus lactis* subsp. lactis, *Leuconostoc* species, *Lactococcus lactis* subsp. lactis biovar. Diacetylactis, and a *Streptococcus thermophilus* strain	Improvement in antioxidants (AOX) content and viscosity	[67]
Red bean beverage	*Streptococcus thermophilus TISTR 894 (ST)*, *Lactobacillus plantarum 299 V*, and *Lactobacillus casei 388,* as a single or a mixed culture fermentation	Improvement in AOX content	[68]
Bean (*Phaseolus vulgaris*) beverage	10 lactobacillus strains	Decrease in saturated fat and increase in unsaturated fat.	[72]
Barley:finger millet: moth bean	*Lactobacilli acidophilus* and *a probiotic bacterium*	Increase in polyphenol content	[52]
Chickpea beverage	*Streptococcus thermophilus* (ST), a co-culture of ST with *Lactococcus lactis* and a co-culture of ST with *Lactobacillus plantarum*	Decrease in saturated fat, phytic acids and increase in minerals	[19]
Soymilk	*Lactobacillus casei PLA5*	Increase in β-glucosidase, minerals and AOX activity and decrease in polyphenols content	[71]
Bean (*Phaseolus vulgaris*)	*Streptococcus thermophilus* + *Lactobacillus Bulgaricus subs Debulgaricus*, *Lactobacillus acidophilus La-5* + *Bifidobacterium animalis Bb-12* + *Streptococcus thermophilus*, *Lactobacillus rhamnosus yoba* + *Streptococcus thermophilus and Fiti*, *Lactobacillus rhamnosus GR1* + *Streptococcus thermophilus*	Increase in B vitamins and decrease in verbascose, stachyose and raffinose	[73]
Soymilk	*Lactobacillus rhamnosus* and *Lactobacillus casei*	Increase in β-glucosidase activity and aglycones	[75]
Black soybean beverage	*Lactiplantibacillus plantarum WGK 4*, *Streptococcus thermophilus Dad 11*, and *Lactiplantibacillus plantarum Dad 13*	Increase in AOX activity and aglycone content	[74]
Soymilk	*Enterococcus faecalis VB43*	Reduction in the immunoreactivity of soybean allergens	[80]
Moringa leaves and beetroot extract drink	*Lactobacillus plantarum* and *Enterococcus hirae*	Reduction in reffinose by 60%, increase in antibacterial activity against pathogenes and improvement of radical scavenging activity and phenolic content as well as minerals	[77]
*C. vulgaris* and soy extract	*Lactobacillus fermentum* and *Lactobacillus rhamnosus*	Increase in polyphenol content and dietary antioxidant capacity	[78]
Chickpea beverage	*Lactiplantibacillus plantarum* subsp. *plantarum*	Reduction in the immunoreactivity of chickpeas proteins	[81]
Soymilk	*Lycoperdon pyriforme*	Decrease in the green off-flavor	[84]
Pea protein isolate drink	*Lactobacillus Plantarum*	Reduction in the off-flavor VOC (aldehydes and ketones)	[87]
Soybean beverage	*Naematelia aurantialba*	Increase in AOX activity, nutrient content and decrease in the oddly flavored VOC	[86]
Oat-based “yoghurt”	*S. thermophilus* and *L. delbrueckii* subsp. *Bulgaricus*	Improvement in the texture and flavor	[94]
Quinoa-based “yoghurt”	*Weissella confusa*	Improvement in the viscosity	[102]
Soy-based “cheese“	*Lactic bacteria* and/or *Geotrichum candidum*	Improvement in the sensorial properties	[116]
Pea protein isolate “cheese”	*Lactobacillus plantarum*, *perolens*, *fermentum*, *casei*, *Leuconostoc mesenteroids* subsp. *Cremoris* and *Pedicoccus pentasaceus*	Increase in cheesy aroma, acid and salty and reduction in the immunoreactivity of allergenic proteins	[64]
Flaxseed oil-based “cheese”	*Penicillium camemberti* and *Geotrichum candidum*	Production of camembert-like cheese with good oil oxidative stability	[119]
Cashew-based “cheese”	Quinoa starter inoculum (dominated by *Pediococcus* and *Weissella*)	Reduction in allergenicity associated with cashew and increase in the viable bacterial count	[120]

## 4. Impact of Fermentation on the Quality of Plant-Based Meat

In conventional fermented meat products, lactic bacteria (*Staphylococcus carnosus* and *Staphylococcus xylosus*) can accelerate the degradation of proteins and fats to produce flavor compounds, enhance palatability, develop compact meat quality, and extend shelf-life (by inhibiting the growth of food-borne pathogenic bacteria such as *Escherichia coli* and *Enterobacteriaceae*) [121]. The most used lactic bacteria were *Lactobacillus* spp., such as *L. plantarum*, *L. sake*, *L. paracasei*, and *L. fermentum* [122]. Lactic acid bacteria were also used for the fermentation of plant-based protein ingredients to produce plant-based meat alternatives. These bacteria (*Lactobacillus *plantarum** P1, *Lactobacillus brevis* R, *Lactobacillus acidophilus* 336, and *Lactobacillus acidophilus* 308) improved water/oil holding capacities and reduced protein oxidation of soy press cake [123]. The inclusion of fermented soy products (10%) in meat alternative formulations improved texture (by increasing juiciness) and flavor (by reducing bitterness and balancing taste) [123]. Edible fungi species (*Lentinus edodes*, *Coprinus comatus* and *Pleurotus ostreatus*) were used as ingredients in making fermented sausages. Among the different species, extruded *Coprinus comatus* and soybean protein showed improved functionality and fibrous structure. The resulting meat alternatives had desirable physicochemical and textural properties, taste, and flavor. Regarding the aroma profile, the curing and fermenting process contributed to the increased volatile compounds’ contents, while fermented sausages without curing showed undesired flavors [124]. Textured vegetable proteins (made by extruding soy protein, corn starch and wheat gluten) were fermented using *B. subtilis*, which resulted in improving the chewiness, hardness, integrity index, and layered structure [125].

Biomass fermentation was used in producing Quorn™ brand products (http://www.quorn.com/ accessed on 20 September 2022). The process relied on a continuous fermentation of glucose from roasted barley malt and nitrogen from ammonia by an edible fungi (*Fusarium venenatum*) [126]. Mycoprotein exhibited organoleptic properties resembling meat, but with longer shelf-life and lower fat content [127]. Furthermore, mycoprotein contains all essential amino acids, and has a protein digestibility corrected amino acid score (PDCAAS) of 0.99 [128].

Yeast biomass could be incorporated into alternative meat formulations to improve their flavor [129]. Particularly, yeast extracts from *Saccharomyces cerevisiae* have been widely used as flavoring agents in many meat products [130]. They are also commonly used in plant-based meat products to impart meat flavor and umami taste. Commercial yeast extract products include Marmite^®^ and Vegemite^®^, which are by-products of fermentation. Torula yeast (*Candida utilis*), obtained using a continuous fermentation process, can be also added as a flavoring agent owing to its natural smoky umami flavor.

Precision fermentation enabled the production of targeted ingredients that could benefit the mimicking of meat products. Soy leghemoglobin has been produced by an engineered yeast *Pichia pastoris* to give the flavor and color of animal meat to plant-based burgers (Impossible Foods) [131]. Since it is a genetically edited ingredient, there were considerable concerns over its safety [132]. A recent metanalysis of the literature showed that foods containing recombinant soy leghemoglobin are unlikely to present an unacceptable risk of allergenicity or toxicity to consumers [133]. Hemami™ (Motif FoodWorks, Boston, MA, USA) is another yeast-derived heme protein. This product delivers an umami flavor and meaty aroma which likely can improve consumers’ sensory perception of plant-based products. Precision fermentation is also used to make fats with similar molecular structures as their animal-derived counterparts [134]. Melt & Marble has developed a precision fermentation-derived beef fat alternative via yeasts. Significant amounts of vitamin B12 were produced using co-fermentation of *Propionibacterium freudenreichii* and *Lactobacillus brevis* in wheat bran [135,136]. This vitamin can be used to fortify plant-based meat products since it is naturally absent in plant-based sources and exclusively found in animal products. Precision fermentation-derived enzymes could be used as post-processors to address the functional limitations of plant proteins. Established enzymes manufacturers can lead this sector through using their expertise to develop a custom portfolio of enzymes able to overcome plant protein challenges. Computational biology “omics” and process engineering would contribute to screening and identifying the potential existing variants and designing new variants with new functionalities.

## 5. Role of Digitalization in the Innovation of Fermented Plant-Based Dairy and Meat Alternative Products

As the plant-based market expands, so does progress in innovation and technological advances in many food-related sectors. Recent technological innovations have been driven by the emergence of the fourth industrial revolution (Industry 4.0) and its advanced technologies such as artificial intelligence (AI), big data, and the Internet of Things (IoT), among others [137,138]. Many publications have shown that digital technologies and other Industry 4.0 innovations could provide tremendous opportunities to improve food quality and traceability, and boost food sustainability [139,140,141].

Attempts to create innovative solutions for healthier diets with alternative proteins have been accelerated with the advent of digital technologies and other advanced related innovations. It was reported that the application of digital technologies, such as AI, smart sensors, robotics, and augmented reality, during fermentation can improve the monitoring and performance of the process [142]. For example, throughout the fermentation process of rice wine, multiple parameters such as temperature, humidity, percentage of sugar and alcohol, and acidity can be measured using IoT, allowing manufacturers to virtually monitor the whole fermentation process online [143]. Incorporation of Industry 4.0 technologies into fermentation facilities, “Fermentation 4.0” has recently been discussed, highlighting its potential to solve relevant problems such as the implementation of complex culture conditions [144]. The implementation of such advanced technologies to achieve automatic detection and control of beer fermentation was recently reviewed and thoroughly discussed [145]. These technologies will enable a better understanding of the fermentation process (classic/biomass) and thus an advanced control of the process to reach desired texture/viscosity.

Sophisticated and automated processes, such as automated computer vision would help in modulating process conditions to maximize the production and the quality. Monitoring the process will offer a better understanding of the synergy among the bacteria/fungi used and how they interact within different media (different raw materials unlike milk). Three-dimensional (3D) cameras and hyperspectral imaging (HSI) would enable real-time monitoring, and thus the process can be adjusted and optimized in real-time during processing [146,147]. For example, in a recent study, HSI was used to predict and quantify the total acid content and reducing sugar content of fermented grains [148]. The results showed that HSI can be used to monitor the fermented grains’ process in a rapid and non-destructive manner compared to traditional methods.

Fermentation contributes to improving flavor through the formation of volatile compounds, thus advanced analytical tools for their identification/quantification are deemed crucial to modulate the process/ingredients/microorganisms to reach desirable sensory properties such as aroma and flavor. To this end, omics technologies and bioinformatics tools, in addition to AI and big data, are being increasingly investigated [140]. Omics techniques applied during the production of vegetable-fermented foods and beverages could help us to determine and quantify microbial composition, understand the metabolic and functional properties of the microbial communities, detect changes associated with their development, and identify the metabolites that they produce [149]. High-throughput analytical techniques, such as advanced mass spectrometry, chromatography, and nuclear magnetic resonance spectroscopy, are being used to select optimum fermentation conditions, measuring all the enzymes and metabolites produced by the microbes [140,150].

The application of digital technologies and other advanced innovations has been demonstrably efficient in upgrading the quality and safety of fermented foods and beverages [151,152]. Nevertheless, similar applications could be expected in the near future in the sector of fermented plant-based dairy and meat alternative products. Recent advances in Industry 4.0 technologies have enabled vast progress in precision fermentation due to recent advances in AI, bioinformatics, and systems and computational biology [134,153]. Precision fermentation scales up the production of plant-based ingredients that will require advanced technologies for monitoring and optimizing the process. Such advances could offer quality standardization by detecting any potential anomalies in the fermentation process (e.g., mutation), and stable productivity, and thus could be more cost-effective food production methods. Advances in precision fermentation are expected to be key elements in the future to target taste and texture, enhance the shelf-life of plant-based fermented food products, and to mimic their animal counterparts [154,155,156].

Additive manufacturing (or 3D printing) is among the emerging technological alternatives to traditional food production methods. 3D food printing and its derivatives (i.e., 4D, 5D, and 6D printing) have experienced a rapid evolution over the last few years, revolutionizing many aspects of the food industry [157,158,159,160]. According to our search inquiry on the Scopus database, there has been a significant increase in the number of publications and citations reporting on the application of 3D printing in the food sector over the last decade (Figure 3). 3D food printing integrates digital gastronomy with additive manufacturing technology [161,162]. Applications of 3D printing in the food sector are gaining increasing relevance due to their high potential in the production of personalized food, the reduction of food waste, and time and energy savings among other benefits [162,163,164].

The combination of 3D food printing with AI can increase exploration of novel protein sources from plants, insects, fungi, and algae [165]. AI and machine learning can be applied to perform a thorough analysis of ingredients that can be used in the plant-based industry to produce alternative products with similar molecular structure, taste, and texture to products of animal origin [166,167]. A wide variety of plant-based materials have been proven to be suitable for 3D food printing [168]. For example, a recent study investigated the possibility of producing hybrid meat analogues prepared by printing pea protein isolate and chicken mince as plant and animal protein ingredients at different ratios [169]. Although there is a transition toward more plant-based diets being more sustainable and environmentally friendly, the question of the ability of such a diet to fulfill nutritional requirements such as the composition of essential amino acids in proteins is still under debate. In a recent study, Conzuelo and co-authors developed a digital tool that can provide a combination of protein ingredients (e.g., soy products, microalgae, and press cakes) used in the formulation of nutritious dairy/meat analogues and snacks with a high-to-excellent protein quality [170]. However, the organoleptic attributes and technological performance of the protein combinations were not addressed in this study.

The use of 3D printing as a post-processing technique of fermented plant-based products might boost their quality to a higher level in terms of nutrition, taste, aspect, and color, as well as product stability. It was reported that 3D printing of processed dairy cheese enabled higher dimensional stability, color, casein retention rate, and a final porosity compared to traditional methods [171]. The technique could be applied to make plant-based fermented “cheese” alternatives to guarantee a better structure, color, and taste. For instance, making plant-based hard “cheese” is still a challenging task, and thus the use of 3D printing might give the possibility of creating “cheese” alternatives with complex geometries. Similarly, extrusion-based 3D printing of meat alternatives can make 3D structures similar to animal products. Bacterial species are now being mixed with various bioinks to produce functional complex materials using 3D printing [172]. These systems could use fermentation substrates to make probiotic products with tailored features to facilitate personalized nutrition [173]. Further optimization of this technique would enable the production of novel 3D-printed fermented meat and dairy alternative products with improved nutritional and functional properties. Thus, combining 3D-food printing technology and biotechnology approaches would help us to achieve products with equivalent taste, nutrition, structure and flavor to those of animal products [174]. As 3D printing is well adapted to create new foods and textures, it is expected that this revolutionary technology will be the next big thing in the next decades. Finding a solution to overcome the limitations of 3D printing technology, such as consumer acceptance, insufficient institutionalization, and lack of standardized food material, would boost its use [174,175].

To summarize, the application of digital technologies in fermented plant-based dairy and meat alternative products is currently limited, but further innovations are expected to accelerate the development of these products.

## 6. Conclusions

Fermentation is attracting plenty of attention as a green sustainable solution to design plant-based alternatives with similar features to those of traditional foods. The key advantages of traditional and biomass fermentation are the familiarity and versatility of microorganisms. These microorganisms are also of natural origin and sustainably processed, providing minimally processed ingredients/biomass/products with enhanced organoleptic properties and health benefits. The addition of probiotics to the manufacture of plant-based dairy alternatives ensured improved nutritional and organoleptic values and an improved shelf-life of the products (beverages, yoghurt, and cheese). More research is required to select adequate starter suitable for fermenting plant-based material to offer nutritionally balanced products with desirable taste and flavor.

Precision fermentation could deliver targeted compounds that might upgrade the quality of plant-based foods to match that of animal products. However, its main challenges include consumer perception of genetically engineered products, scalability, and ethical and regulatory concerns. Consumer understanding of precision fermentation is still lacking, and thus raising awareness about innovative food technologies is required to avoid the gap between consumers and the science behind their foods. Close collaboration between different actors in the food value chain is needed to overcome other obstacles. In the future, qualitative and quantitative sensorial studies are required for a better understanding of consumer acceptance/rejection of precision fermentation. The digestibility of plant-based fermented meat and dairy products and its effects on the composition of gut microbiota are scarcely investigated. These studies are of high relevance for understanding the impact of fermentation on human health. More research is also required to harness the opportunities offered by digitalization and other Industry 4.0 technologies for healthier and more sustainable food.

## Figures and Tables

**Figure 1 foods-12-01005-f001:**
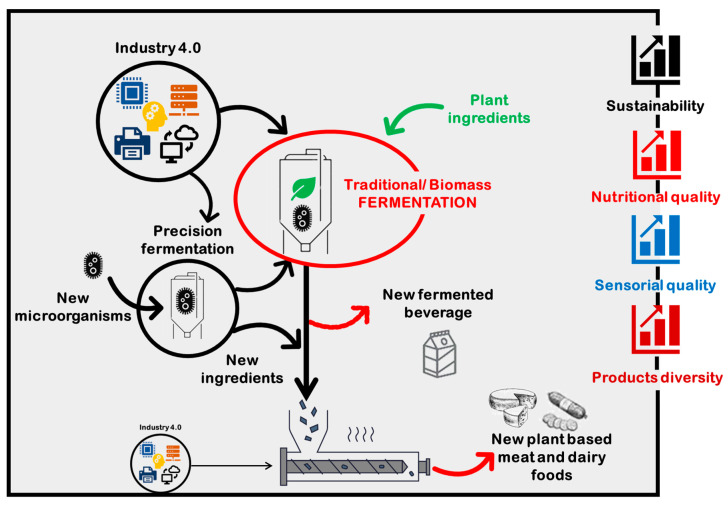
The use of fermentation to make innovative plant-based products.

**Figure 2 foods-12-01005-f002:**
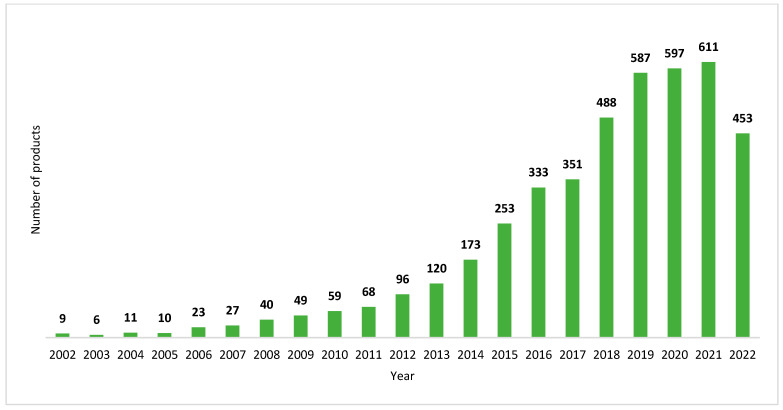
Evolution of launches of fermented plant-based foods in the global market. A search was conducted on Mintel GNPD (Global New Products Database) on 15 September 2022. Date: from 1 January 2002, to 15 September 2022. The inclusion criteria of the search were as follows: the Super-Category matches Food and Drink, the Claims match one or more of [Vegan/No Animal Ingredients; Plant Based], and Ingredient Search matches Fermented as one of the Ingredients. A total of 4379 products were retrieved.

**Figure 3 foods-12-01005-f003:**
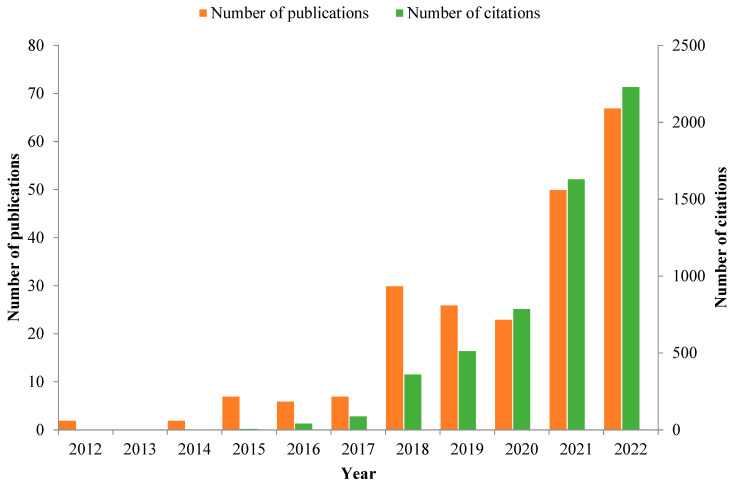
Number of publications and citations/year on 3D food printing over the last decade. The search was done on 1 November 2022, using the following search query criteria on the Scopus database: TITLE (3D food printing) OR (Additive food manufacturing).

**Table 1 foods-12-01005-t001:** Fermented plant-based foods sold in the global market.

Fermented Plant-Based Foods and Drinks	Number of Products and Percentage ^1^
Foods	
Bakery	1371 (31.2%)
Dairy alternatives	1152 (26.3%)
● “Yoghurt”	420 (9.6%)
● Hard “cheese” and semi-hard “cheese”	210 (4.8%)
● Soft “cheese” and semi-soft “cheese”	188 (4.3%)
● Processed “cheese”	141 (3.3%)
● Ice cream	100 (2.3%)
● Drinks	50 (1.1%)
● Fresh cheese and cream cheese	35 (1%)
● Margarine	8 (0.18%)
Sauces and seasonings	467 (10.2%)
Meat alternatives	378 (8.6%)
Ready-to-eat meals	312 (7.1%)
Vegetables	93 (2.1%)
Drinks	
Nutritional drinks and other beverages	260 (5.9%)
Carbonated soft drinks	109 (2.5%)
Alcoholic beverages	40 (0.9%)
Juice drinks	39 (0.9%)
Sports and energy drinks	8 (0.2%)
Ready-to-drink beverages	4 (0.1%)

^1^ Total of fermented plant-based products = 4379. A search was conducted on Mintel GNPD (Global New Products Database) on 15 September 2022. Date: from 1 January 1996, to 15 September 2022. The inclusion criteria of the search were as follows: the Super-Category matches Food and Drink, the Claims match one or more of [Vegan/No Animal Ingredients; Plant Based], and Ingredient Search matches Fermented as one of the Ingredients. A total of 4379 products were retrieved.

**Table 2 foods-12-01005-t002:** Landscape of the global market of fermented plant-based foods.

Region	Number (% of Total) ^1^	Top 10 Brands
Europe	3128 (71%)	Fentimans: Alpro; Tesco Finest; M & S Food; M & S The Bakery; Tesco; Asda Extra Special; BFree; Marks & Spencer; Sojasun
Asia Pacific	729 (17%)	Maggi; Pascual; East Bali Cashews; Javara; Lo Bros.; Prima Ham Try Veggie; Remedy Kombucha, Coles; Elle & Vire; Fentimans
North America	299 (7%)	Genuine Health; Field Roast Chao; Nuts for Cheese Naked & Saucy; BFree; Field Roast Chao Vegan Creamery; Health-Ade Pop; Booch; Brami; Hu
Latin America	154 (4%)	Nomoo; Ile de France; Milkaut; Liane; Mun; Augusta; Emporium Vida; Neptune; Nogurt; Soignon
Middle East & Africa	69 (2%)	Woolworths Food; Soignon; Carrefour; Fry’s Special Vegetarian; Kefir Life; Moya; Vigo Kombucha; Woolworths; Fynbos Fine Foods; Herman Brot

^1^ Total of fermented plant-based products = 4379. A search was conducted on Mintel GNPD (Global New Products Database) on 15 September 2022. Date: from 1 January 1996, to 15 September 2022. The inclusion criteria of the search were as follows: the Super-Category matches Food and Drink, the Claims match one or more of [Vegan/No Animal Ingredients; Plant Based], and Ingredient Search matches Fermented as one of the Ingredients. A total of 4379 products were retrieved.

## Data Availability

Not applicable.

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
