# Peer review of "Fermentation for Designing Innovative Plant-Based Meat and Dairy Alternatives"

_foods, 2023, doi:10.3390/foods12051005_

Round 1

Reviewer 1 Report

This review is very interesting and offers new insights into the possibilities of fermentation in the production of dairy and meat alternatives. The paper need remove the statements that are not supported by the results of independent scientific studies. In addition, some minor corrections should be made according to the comments included in the manuscript.

Author Response

This review is very interesting and offers new insights into the possibilities of fermentation in the production of dairy and meat alternatives. The paper need remove the statements that are not supported by the results of independent scientific studies. In addition, some minor corrections should be made according to the comments included in the manuscript.

The authors thank the reviewer for the positive feedback and the valuable comments.

Page 2

As part of.......around the world.

The sentence was rephrased in Line 45-46.

That's a little ambiguous. I assume you meant to compare plant and animal sources nutritional quality in general, not plant sources with animal proteins specifically? Or you are saying that plant sources have a lower nutritional quality than animal proteins, which is of course true?

The sentence was modified to avoid the ambiguity as follows:

These plant-based sources have lower nutritional quality than that of animal proteins and might induce undesirable off-flavors. (line 50-51)

Please specify biomass.

"...to obtain single cell proteins (SCP)." There is no need for repeating "biomass"

The sentence was modified as follows:

“Biomass fermentation aims to obtain single cell proteins (SCP).” (line 80)

This statement is too general to be supported by only one (self-) citation, i.e. the results of a single research study on a specific type of waste. The claim in this form should be confirmed by additional recent references

More references were added to support this statement. (line 89)

Page 3

These are sweeping claims that are not clearly and fully based on scientific research, but rather the results of a current global media campaign driven by interest groups and policy makers, rarely by the consumers themselves, in general. In addition to the claims about the environmental and health benefits of plant-based diets, some of which are undoubtedly correct, there are other disadvantages apart from their lower nutritional value and poorer structural properties that should be mentioned; notably the significantly higher retail prices of such ready-made products compared to conventional dairy and meat products which makes them practically unavailable to average consumer. Not to mention that intensive production of cereals is also a significant source of greenhouse gas (GHG) emissions from the production of synthetic nitrogen (N) fertilizers consumed in crop production.

This statement was removed to avoid any misunderstanding.

Page 4

Please specify the period of the increase

The sentence was clarified as follows:

In 2020, consumer interest in vegan (1.9%), vegetarian (3.1%), and flexitarian (57.1%) increased across Europe (line 131).

Please rewrite this sentence as follows: Plant-based dairy and meat alternatives produced by fermentation account for 26.3% and 8.56%, respectively, of total fermented plant-based foods sold on the market.

The sentence was rewritten as suggested by the reviewer.

Dairy terms - milk, cheese, butter, and yogurt - are protected and must only refer to products derived from animals (with some exceptions e.g. butterbeans, peanut butter, custard creams). So when you write about plant-based product similar to yogurt than you should use hyphens: "yogurt".

The designation was corrected throughout the manuscript.

Page 8

The term is incorrect, these are not cheeses but cheese analogues, or plant-based substitutes/imitations for/of cheese

Corrections were made throughout the manuscript.

Please rewrite this sentence as follows: "The use of plant-based fermentation in cheese is not yet widespread on the market."

The sentence was rewritten as suggested by the reviewer. (line 289)

Page 10

This is partially correct. Indeed, Staphylococcus carnosus and Staphylococcus xylosus can promote the hydrolysis of fat and protein in preserved meat, improve color and assist in the rapid formation of flavor. The main role of lactic acid bacteria in meat processing, especially in the production of sausages, is to lower the pH by breaking down the carbohydrates in the raw sausage to produce a large amount of lactic acid to inhibit or kill spoilage microorganisms and pathogenic microorganisms. In addition, the lactic acid bacteria control the growth of total bacterial colonies during sausage fermentation and inhibit the growth of food-borne pathogenic bacteria such as Escherichia coli  and Enterobacteriaceae.

The sentence was enriched as follows:

In conventional fermented meat products, lactic bacteria (Staphylococcus carnosus and Staphylococcus xylosus) can accelerate the degradation of proteins and fats to produce flavor compounds, enhance palatability, and develop compact meat quality, clear color, and extend shelf-life (and inhibit the growth of food-borne pathogenic bacteria such as Escherichia coli  and Enterobacteriaceae) [121]. (line 320-324)

This statement should be supported by sound scientific proof. Otherwise it should be deleted. These references do not support the previous statement.

The sentence was removed.

Page 13

Suggestion for this sentence:

They are also of natural origin and sustainably processed, providing minimally processed ingredients/biomass/products with enhanced organoleptic properties and health benefits.

The sentence was rewritten as suggested by the reviewer. (line 494-495)

Reviewer 2 Report

The authors presented in their manuscript a comprehensive and interesting study on the use of fermentation as a pathway to increase the beneficial characteristics of plan-based products as dairy and meat alternatives. Several minor issues should be improved, namely:

1. Please check that wherever italics is used it is needed, in addition please check spaces in the whole text.

2. Should processes that use only enzymes be called fermentation? It seems that the term biotransformation is more appropriate.

3. Page 6 - Plant-based substrates favor high viability of fermenting microorganisms, and consequently result in probiotic dairy-free products (>6 CFU.mL-1 of lactic acid bacteria cocci - it should be 10^6 CFU/mL

4. Please unify - 3 versions: CFU, cfu, colony forming units - can be found in manuscript, the same for mL-1 and /mL

5. Page 6 - Bifidobacteria instead of bifidobacteria

Author Response

The authors presented in their manuscript a comprehensive and interesting study on the use of fermentation as a pathway to increase the beneficial characteristics of plan-based products as dairy and meat alternatives. Several minor issues should be improved, namely:

The authors thank the reviewer for the positive feedback and the valuable comments.

Please check that wherever italics is used it is needed, in addition please check spaces in the whole text.

The text was carefully checked, and changes were made.

Should processes that use only enzymes be called fermentation? It seems that the term biotransformation is more appropriate.

The authors meant with “enzymes”, those produced by microorganisms during fermentation, and not added enzymes.

Page 6 - Plant-based substrates favor high viability of fermenting microorganisms, and consequently result in probiotic dairy-free products (>6 CFU.mL-1 of lactic acid bacteria cocci - it should be 10^6 CFU/mL

Changes were made in the text (line 211)

Please unify - 3 versions: CFU, cfu, colony forming units - can be found in manuscript, the same for mL-1 and /Ml

The text was carefully checked, and changes were made.

Page 6 - Bifidobacteria instead of bifidobacterial

Changes were made in line 217.

Reviewer 3 Report

The paper has an interesting topic and is well-written. 

To improve the paper, I recommend adding some information to each subchapter (beverages, cheese, yogurt-like products, meat) (a paragraph) regarding the addition of other plant-based powders/extracts in the fermentation process (i.e., https://doi.org/10.1016/j.lwt.2015.04.023, https://doi.org/10.1016/j.lwt.2018.11.010, https://doi.org/10.3390/biom13020245, etc.)

Figure 2: in what databases was the research conducted? Please specify. Same for tables 1 and 2. Figures and tables should be stand-alone.

Please place the tables near where they are mentioned in the text. 

Page 5, GIF database- please specify the last time it was accessed.

Table 5: what is the difference between soy drink and soy milk? I recommend keeping the terms consistent throughout the manuscript

 I appreciate section 5!

References 61 and 67 are not according to the journal requirements.

Author Response

The paper has an interesting topic and is well-written. 

The authors thank the reviewer for the positive feedback and the valuable comments.

To improve the paper, I recommend adding some information to each subchapter (beverages, cheese, yogurt-like products, meat) (a paragraph) regarding the addition of other plant-based powders/extracts in the fermentation process (i.e., https://doi.org/10.1016/j.lwt.2015.04.023, https://doi.org/10.1016/j.lwt.2018.11.010, https://doi.org/10.3390/biom13020245, etc.)

More information was added to the text.

Figure 2: in what databases was the research conducted? Please specify. Same for tables 1 and 2. Figures and tables should be stand-alone.

Mintel GNPD (Global New Products Database) was used. This information was added to the manuscript.

Please place the tables near where they are mentioned in the text. 

Tables were placed near the text.

Page 5, GIF database- please specify the last time it was accessed.

The sate was added in line 173.

Table 3: what is the difference between soy drink and soy milk? I recommend keeping the terms consistent throughout the manuscript

Changes were made in table 3.

 I appreciate section 5!

Thank you.

References 61 and 67 are not according to the journal requirements.

References were carefully checked.

Reviewer 4 Report

Fermentation for designing innovative plant-based meat and dairy alternatives.

The review is well written and easy to understand.

I advise the authors to reread the manuscript carefully to correct some language errors and style changes in the writing, for example in paragraph 3.2 and 4.

The focus of the review is certainly important as food choices are changing year by year, moving more towards the sustainable and fermentation is likely to be the way to go.

I would have appreciated it if the paragraph on the nutritional changes that fermentation can bring to food, for example in vitamin or phenolic quantities, was better developed, instead of just citing an article. There is a lot of talk about improving texture and flavour but perhaps these are secondary aspects.

I find the work in any case very interesting .

Author Response

The review is well written and easy to understand.

The authors thank the reviewer for the positive feedback and the valuable comments.

I advise the authors to reread the manuscript carefully to correct some language errors and style changes in the writing, for example in paragraph 3.2 and 4.

The manuscript was carefully checked.

The focus of the review is certainly important as food choices are changing year by year, moving more towards the sustainable and fermentation is likely to be the way to go.

The authors thank the reviewer for the observation.

I would have appreciated it if the paragraph on the nutritional changes that fermentation can bring to food, for example in vitamin or phenolic quantities, was better developed, instead of just citing an article. There is a lot of talk about improving texture and flavour but perhaps these are secondary aspects.

More clarifications about the nutritional quality were added to the manuscript.

I find the work in any case very interesting.

Thank you.